# Deciphering the Gut–Liver Axis: A Comprehensive Scientific Review of Non-Alcoholic Fatty Liver Disease

Samradhi Singh [1,†], Mona Kriti [1,†], Roberto Catanzaro [2], Francesco Marotta [3], Mustafa Malvi [4], Ajay Jain [4], Vinod Verma [5], Ravinder Nagpal [6], Rajnarayan Tiwari [1] and Manoj Kumar [1,*]

[1] ICMR-National Institute for Research in Environmental Health, Bhopal Bypass Road, Bhauri, Bhopal 462030, India; sammradhisingh@gmail.com (S.S.); mkriti2858@gmail.com (M.K.); tiwari.rr@gov.in (R.T.)

[2] Internal Medicine Unit, Department of Clinical and Experimental Medicine, Gastroenterology and Hepatology Service, University Hospital Policlinico "G. Rodolico", University of Catania, 95123 Catania, Italy; rcatanza@unict.it

[3] ReGenera Research Group for Aging-Intervention, 20144 Milano, Italy; fmarchimede@libero.it

[4] Choithram Hospital and Research Centre Indore, Indore 452014, India; malvisilverline@gmail.com (M.M.); ajayvjain@yahoo.com (A.J.)

[5] Stem Cell Research Centre, Department of Hematology, Sanjay Gandhi Post-Graduate Institute of Medical Sciences, Lucknow 226014, India; vverma29@gmail.com

[6] Department of Nutrition & Integrative Physiology, College of Health & Human Sciences, Florida State University, Tallahassee, FL 32306, USA; rnagpal@fsu.edu

* Correspondence: manoj15ndri@gmail.com

† These authors contributed equally to this work.

**Abstract:** Non-alcoholic fatty liver disease (NAFLD) has emerged as a significant global health issue. The condition is closely linked to metabolic dysfunctions such as obesity and type 2 diabetes. The gut–liver axis, a bidirectional communication pathway between the liver and the gut, plays a crucial role in the pathogenesis of NAFLD. This review delves into the mechanisms underlying the gut–liver axis, exploring the influence of gut microbiota, intestinal permeability, and inflammatory pathways. This review also explores the potential therapeutic strategies centered on modulating gut microbiota such as fecal microbiota transplantation; phage therapy; and the use of specific probiotics, prebiotics, and postbiotics in managing NAFLD. By understanding these interactions, we can better comprehend the development and advancement of NAFLD and identify potential therapeutic targets.

**Keywords:** metabolic disorder; NAFLD; gut–liver axis; gut microbiome; hepatic steatosis; probiotics

## 1. Introduction

Non-alcoholic fatty liver disease (NAFLD) is defined by the presence of at least 5% hepatic steatosis without common secondary causes like excessive alcohol consumption, chronic viral hepatitis, autoimmune hepatitis, congenital hepatic disorders, or long-term use of steatosis-inducing medications [1]. The prevalence of NAFLD is escalating globally, largely due to the obesity epidemic, and is anticipated to become the primary cause of liver transplantation by 2030, leading to rising healthcare costs [2]. Over the past few decades, the global prevalence of NAFLD has surged by more than 50%, from 25.3% between 1990 and 2006 to 38.0% between 2016 and 2019, reflecting the parallel rise in obesity and type 2 diabetes (T2D) [3]. In the majority of patients, NAFLD is associated with comorbidities like obesity, beta cell dysfunction, insulin resistance, T2D, and dyslipidemia [4]. The stringent associations between NAFLD and its mortality-driving comorbidities are not entirely understood but may involve continuous low-grade inflammation [2].

As our understanding of the metabolic underpinnings of this condition has evolved, new terminology and diagnostic criteria have been proposed, i.e., the term metabolic dysfunction-associated fatty liver disease (MAFLD) was introduced to better capture these metabolic associations. According to a 2020 consensus statement, MAFLD diagnosis

requires the presence of hepatic steatosis in addition to one of the following criteria: overweight/obesity, type 2 diabetes, or evidence of metabolic dysregulation (elevated waist circumference, hypertension, hypertriglyceridemia, low HDL cholesterol, insulin resistance, or elevated C-reactive protein levels) [5,6]. More recently, in 2023, the term metabolic dysfunction-associated steatotic liver disease (MASLD) has been proposed to refine and simplify the diagnostic criteria. MASLD can be diagnosed if hepatic steatosis is present along with one of five cardiovascular risk factors: hypertension, type 2 diabetes, obesity, hypertriglyceridemia, or low HDL cholesterol, and without metabolic risk factors, the condition is termed cryptogenic steatotic liver disease (SLD). Additionally, a new category called metabolic dysfunction and alcoholic liver disease (MetALD) has been introduced for individuals with MASLD who consume more alcohol than the threshold for nonalcoholic status but less than the threshold for alcoholic liver disease (ALD) (average daily 20–50 g for women, 30–60 g for men). This shift from NAFLD to MAFLD and now MASLD reflects an ongoing effort to better characterize and diagnose liver diseases associated with metabolic dysfunction, ensuring a more targeted and precise approach to patient care and research [5,6].

Clinically, NAFLD is asymptomatic and is frequently diagnosed incidentally through abnormal liver enzyme findings or imaging studies by the presence of hepatic steatosis [7]. Additionally, NAFLD is driven by the excessive hepatic accumulation of lipids, particularly free fatty acids, fueled by the non-esterified fatty acid pool, dietary triglycerides, and de novo lipogenesis [8]. Lipotoxicity from non-esterified fatty acids and diacylglycerol promotes hepatic insulin resistance and endoplasmic reticulum stress, leading to chronic inflammation, liver fibrosis, cirrhosis, and ultimately hepatocarcinogenesis. As NAFLD progresses to non-alcoholic steatohepatitis (NASH), it is characterized by hepatic steatosis, inflammation, and cellular damage. Insulin resistance, oxidative stress, and pro-inflammatory cytokines such as TNF-$\alpha$ and IL-6 are key in this progression [9]. Persistent inflammation and liver damage in NASH can result in fibrosis, where extracellular matrix components are deposited in an attempt to repair the liver. Fibrosis advances through stages, ultimately leading to cirrhosis, where extensive scarring disrupts liver architecture and function, significantly increasing the risk of hepatocellular carcinoma (HCC). Mechanisms leading to HCC include genomic instability from chronic inflammation, epigenetic changes, and proliferative signaling due to ongoing liver cell damage and regeneration. Dysregulation in pathways such as insulin/IGF signaling, Wnt/$\beta$-catenin, and p53, as well as angiogenesis facilitated by factors like Vascular endothelial growth factor (VEGF), further contribute to carcinogenesis [9].

Theories on the pathogenesis of NAFLD have evolved from the two-hits hypothesis to the current multiple-hits hypothesis. The two-hits hypothesis posits that the first hit involves triglyceride accumulation in the liver (hepatic steatosis) due to sedentary lifestyles, high-fat diets, insulin resistance, and obesity. This accumulation increases the liver's susceptibility to a second hit, characterized by lipotoxicity from free fatty acids, which then activates pro-inflammatory cytokines, promotes oxidative stress, and triggers fibrogenesis, leading to severe NAFLD phenotypes [10]. However, this hypothesis is now considered outdated as it fails to account for the roles of nutrition, gut microbiota, adipose tissue hormones, and concurrent insulin resistance in genetically predisposed individuals. The multiple-hits hypothesis suggests that genetic factors, diet, and environmental influences can cause gut microbiota dysbiosis, insulin resistance, and obesity, which together promote NAFLD development and are believed to interact in complex, interrelated ways [10]. Recent research has rapidly uncovered the link between gut microbiota and NAFLD, particularly in cases of obesity-related and high-fat-diet-induced NAFLD in both adults and children. Dysbiosis, or disruption of the typically beneficial gut microbiota, promotes the development of NAFLD by altering gut–liver homeostasis. This includes dysregulation of the gut barrier, transport of lipopolysaccharide (LPS) to the liver, altered bile acid profiles, and decreased short-chain fatty acids (SCFAs). The increased recognition of the importance of gut microbiota-mediated homeostasis in preventing NAFLD suggests that

gut microbiota-targeted preventive and therapeutic strategies, such as probiotics, could be effective in combating NAFLD [10–12]. This review explores the intricate interactions and communication between gut microbiota and the liver, proposing potential therapeutic strategies focused on modulating gut microbiota for managing and treating NAFLD.

### 1.1. The Gut–Liver Axis: Key Insights and Interconnections

The gut–liver axis refers to the bidirectional network of signals (Figure 1) between the gut and the liver, connected via the hepatic portal vein, bile tract, and systemic circulation [13]. This axis is crucial for understanding the pathogenesis of various metabolic disorders including NAFLD and its progression to hepatocellular carcinoma (NAFLD-HCC).

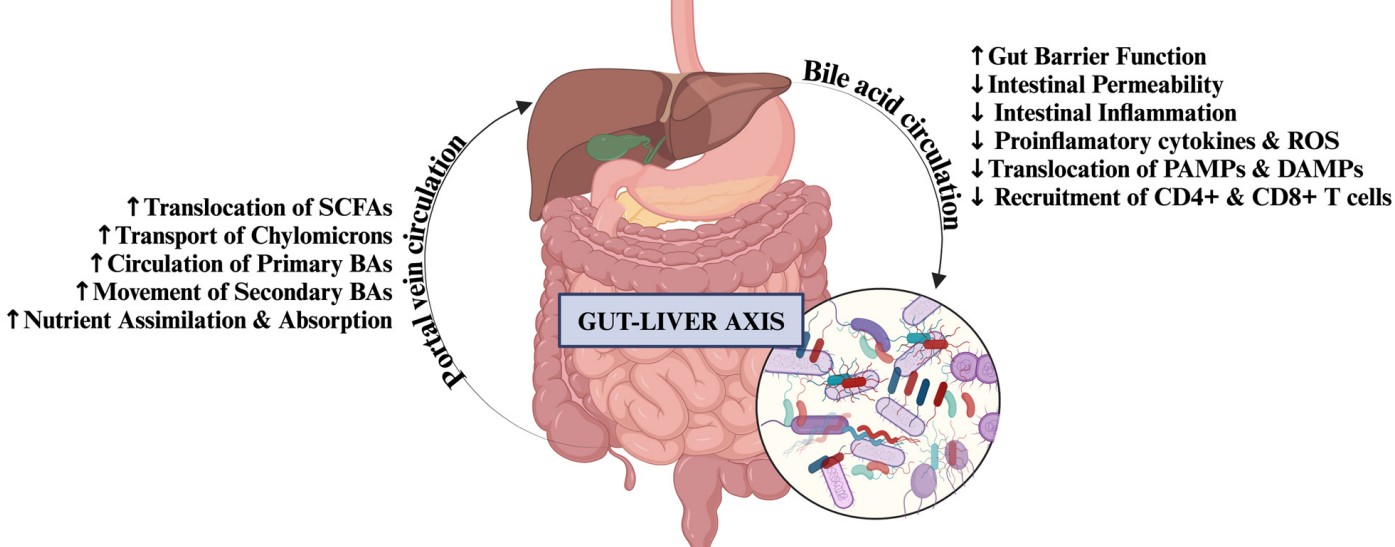

**Figure 1.** This schematic illustrates the complex bidirectional communication between the liver and gut, known as the gut–liver axis. The diagram also highlights the flow of various substances and complex signals through this axis, which plays a crucial role in maintaining homeostasis and influencing disease states. SCFA: Short Chain Fatty Acids; BA: Bile Acids; ROS: Reactive Oxygen Species; PAMPs: Pathogen-Associated Molecular Patterns; DAMPs: Damage-Associated Molecular Patterns; CD4: Cluster of Differentiation.

The liver receives nearly two-thirds of its blood supply from the gastrointestinal tract via the portal vein, which transports nutrients, bacteria, and their components to the liver while bile and antibodies return to the small intestine to regulate the gut microbiome [14–16]. The gut–liver axis plays a crucial role in NAFLD pathogenesis due to its anatomical and functional connections. Dysbiosis, an imbalance in gut microbiota, can worsen the condition by increasing intestinal permeability, allowing harmful bacteria and their components to enter the liver. This stimulates hepatic immune cells and activates inflammatory pathways, potentially leading to NAFLD, which can progress to NASH, liver fibrosis, cirrhosis, and ultimately cancer [1,5,17,18]. This interaction is bidirectional, as gut dysfunction can impact liver health and vice versa, creating a cycle that promotes disease progression [19,20]. The gut–liver axis's role in NAFLD is supported by the multiple-hit hypothesis, which states that different genetic, environmental, and lifestyle factors contribute to the disease development and progression. Other factors such as high caloric intake, physical inactivity, genetic predispositions [21], obesity, and gut dysbiosis-related damage to the intestinal integrity can lead to gut–liver axis malfunction, which might contribute to simple steatosis, thereby allowing bacterial components to enter circulation, a condition known as the "leaky" gut. This permeable gut permits hepatotoxic bacterial substances, including

pathogen-associated molecular patterns (PAMPs) and damage-associated molecular patterns (DAMPs), to reach the liver through the hepatic portal circulation and activate toll-like receptors (TLRs) in hepatic cells. These TLRs sense bacterial products like lipoteichoic acid and lipopolysaccharides (LPS) influencing gut barrier function and permeability [22,23].

Subsequent triggers by increased lipotoxicity and enhanced intestinal permeability further worsen the metabolic and systemic profile of NAFLD patients [21]. Recent research emphasizes the key role of the gut–liver axis in liver diseases beyond NAFLD and states that the disruptions in the gut–liver axis, gut microbiome composition, and epithelial barrier function can increase microbial exposure, microbial translocation, infections, and a pro-inflammatory environment in the liver, thus causing disease progression [22].

### 1.2. Distinct Gut Microbiome Signatures in NAFLD Patients

The intestinal tract harbors a vast number of microorganisms that play a vital role in maintaining metabolic homeostasis [10,24,25]. Among these, *Firmicutes* (primarily Gram-positive Bacilli, *Clostridia,* and *Mollicutes*) and *Bacteroidetes* (including Gram-negative *Prevotella*, *Alistipes*, and *Parabacteroides*) are the most dominant, constituting about 90% of the gut microbiota [19,24,26,27]. The epithelial mucosal and vascular barriers in the gut allow nutrient absorption while preventing the transmission of microbes and their toxins into the circulation. Disruption of gut barriers or dysbiosis, defined as a relative change in the composition and function of an individual's commensal microbiota, can increase liver exposure to harmful substances and create a pro-inflammatory environment [10,26]. In NAFLD patients, gut microbiota imbalances manifest as reduced microbial diversity and an overgrowth of pathogenic bacteria like *Escherichia coli*, impairing the gut microbiota's ability to maintain local homeostasis, whereas healthy individuals typically have higher microbial diversity and lower abundance of pathogenic bacteria [22,28–30] (Figure 2).

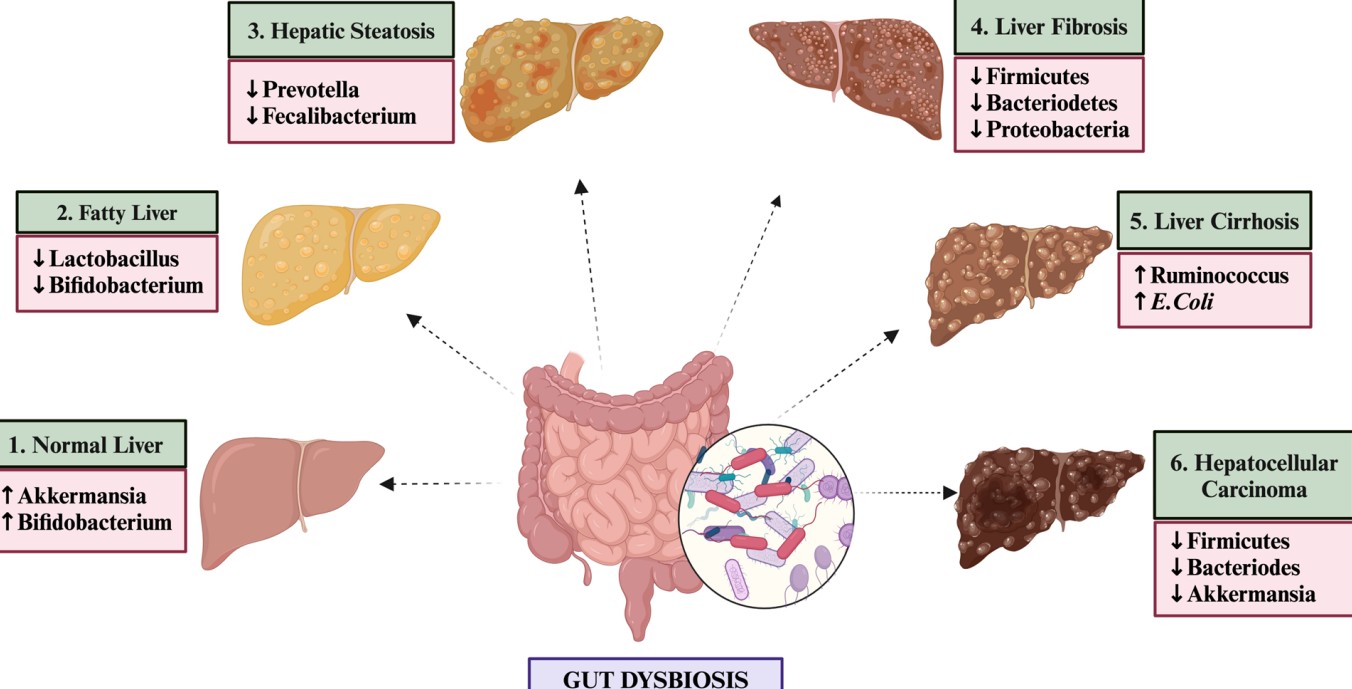

**Figure 2.** This diagram illustrates the progression of Non-Alcoholic Fatty Liver Disease (NAFLD) from a normal liver to hepatocellular carcinoma (HCC), highlighting the critical role of gut dysbiosis which advances through stages like fatty liver, hepatic steatosis, liver fibrosis, and liver cirrhosis. Gut dysbiosis, marked by decreased beneficial bacteria, namely *Firmicutes* and *Bifidobacterium,* and increased harmful bacteria such as *E. coli* and *Proteobacteria*, exacerbates these conditions by promoting inflammation and liver damage.

Among the extensive population of NAFLD patients, an observed increase in *Proteobacteria*, *Lachnospiraceae*, *Escherichia*, *Enterobacteriaceae*, *Barnesiella intestinihominis*, and *Bacteroidetes* is noted, though some studies reported a reduction or no change in *Bacteroidetes* [1,27,31], along with a decline in *Prevotella* and *Firmicutes* [28,29]. Da Silva et al. identified a reduction in levels of certain groups of gut microbiotas, including the *Bacteroidetes* and *Firmicutes* phyla, and an increase in the abundance of another group of microbiota in subjects with simple steatosis and NASH compared to controls suggesting that a particular gut microbiota may play a crucial role in the development and advancement of NAFLD [30]. In a similar study on the Korean population with NAFLD, a relatively lower alpha and beta diversity was reported with an abundance in the phylum *Proteobacteria*, family *Enterobacteriaceae*, genus *Citrobacter* and a significant decrease in the population levels of genus *Faecalibacterium* in subjects with NAFLD when compared to control. Additionally, they also reported a reduction in the abundance of butyrate-producing bacteria and a relative enhancement in the ethanol-producing bacteria in people with NAFLD [32]. In another study, it was revealed that the severity of NAFLD is directly linked to gut dysbiosis and alterations in the metabolic activity of the gastrointestinal microbiota. Here, *Bacteroidetes* was found to be associated with liver steatosis and NASH, while *Ruminococcus* was found to be significantly correlated with fibrosis [33]. A study on 171 Asians with NAFLD revealed significant variations in the microbiome diversity, with *Ruminococcaceae* and *Veillonellaceae* being the primary microbiota associated with the disease [34]. In a similar study on NAFLD patients, the correlation of gut metabolites with the abundance of specific genera was studied, and it was concluded that a decrease in *Oscillospira* is coupled to an up-regulation of 2-butanone, and an increase in *Dorea* and *Ruminococcus* was identified as the gut microbiome signatures linked with NAFLD onset [35].

Studies indicate that NAFLD is associated with a specific disbalance between two predominant phyla, namely *Bacteroides* and *firmicutes*. While some studies suggest increased Firmicutes are linked to NAFLD, others highlight the significant role of Bacteroide overgrowth in disease development [31,36–38]. A study on fecal microbiota of NAFLD cases showed an increased abundance of *Enterobacteriaceae*, *Streptococcus*, *Blautia*, *Flavobacterium*, *Alkaliphilus*, and a significantly reduced level of *Akkermansia* [39,40]. In another study, a substantial rise was found in the *Bacteroidetes/Firmicutes* ratio in NASH subjects independent of diabetes risk factors or drugs such as metformin use [38]. Additionally, NAFLD severity is also linked with increased fecal *Bacteroides* and decreased *Prevotella* levels [33]. Metagenome sequencing shows *Bacteroides vulgatus* and *Eubacterium rectale* are prevalent microbes in mild to moderate NAFLD, while *Escherichia coli* and *Bacteroides vulgatus* dominate in liver fibrosis [36,41].

High-throughput sequencing studies of NAFLD patients have revealed increased levels of *Dysgonomonas*, *Escherichia coli*, *Veillonellaceae*, and *Bilophila*, which promote endotoxin and endogenous ethanol production [19,42]. This leads to elevated systemic inflammation and insulin resistance. Conversely, beneficial bacteria like *Akkermansia muciniphila*, *Faecalibacteriumprausnitzii* (*F. prausnitzii*), *Ruminococcaceae*, *Alistipes*, and *Bifidobacterium* are reduced, impairing SCFA production and compromising the intestinal mucosal barrier [43,44]. Recent research indicates that dysbiosis of *Alistipes* in particular can have both beneficial and detrimental effects, with specific implications in liver fibrosis, colorectal cancer, and cardiovascular diseases. Its distinctive mechanism of fermenting amino acids, termed putrefaction, underscores its significant role in inflammation and liver-related conditions [45–48].

## 1.3. Dynamic Communication Mechanisms between Gut and Liver in NAFLD

The communication between the gut and liver in NAFLD is mediated via several mechanisms including the production and transport of live bacteria and its derived metabolites such as SCFAs, bile acids, trimethylamine oxide (TMAO), ethanol, choline, and amino acids (Figure 3) by the intestinal microbiota and the generation of proinflammatory effects during NAFLD progression [19].

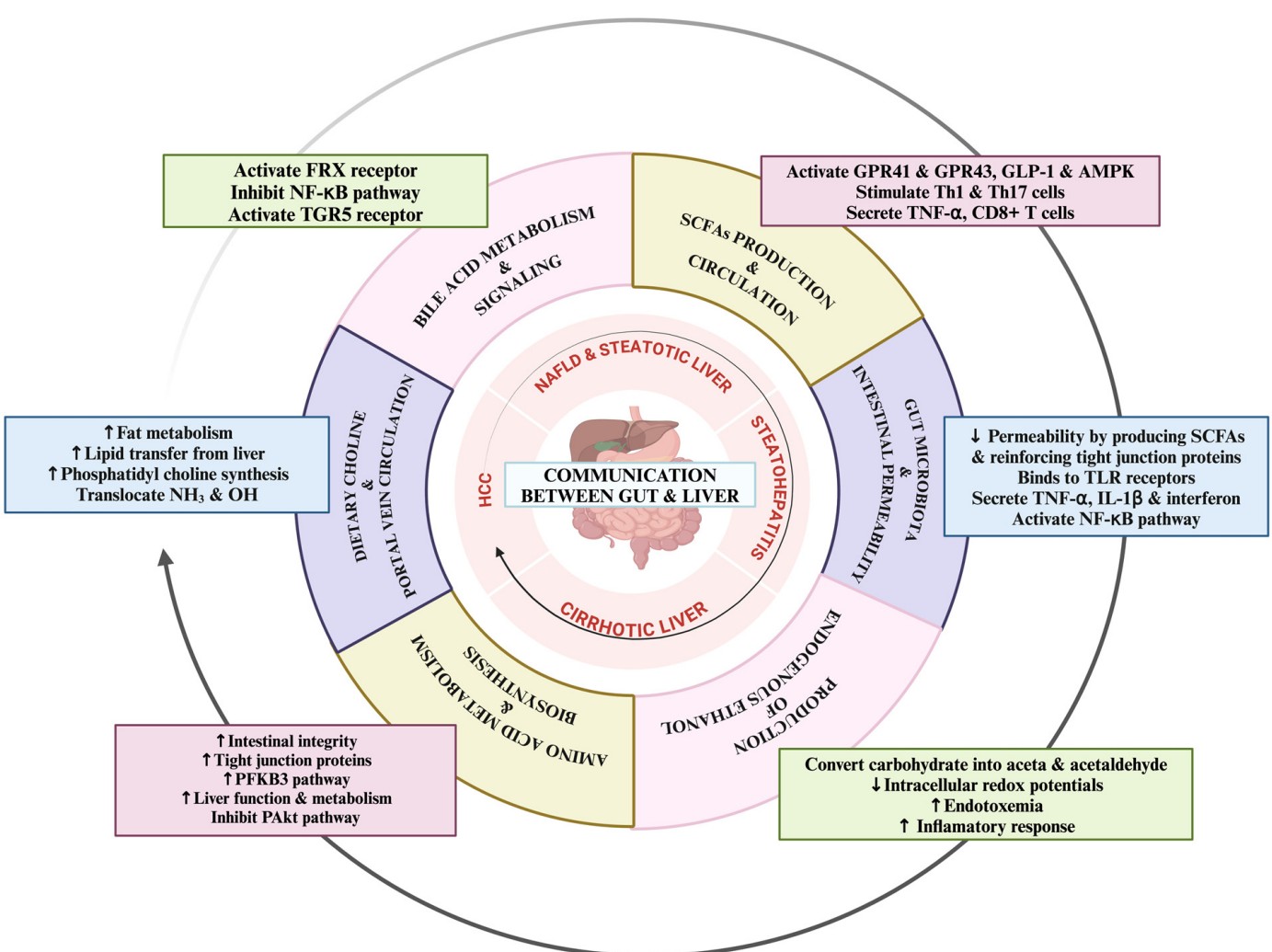

**Figure 3.** The figure illustrates the dynamic interplay and communication mechanisms between the gut microbiota and the liver, highlighting various physiological processes and pathways involved. It also emphasizes how alterations in microbial metabolism and signaling pathways can influence the advancement of NAFLD and its associated complications. FRX: Farnesoid X Receptor; TGR5: Takeda G protein-coupled receptor 5; GPR: G protein Receptors; TNFα: Tumor Necrosis Factor-alpha; Th Cells: T helper cells; AMPK: Adenosine monophosphate-activated protein kinase; CD: Cluster of Differentiation; NH3: Ammonia; OH: Ethanol; PFKB: Phosphofructokinase B; IL1β: Interleukin 1β; NFκB: Nuclear Factor kappa-light-chain-enhancer of activated B cells.

## 2. Short-Chain Fatty Acids: Production, Circulation, and Effects on Liver Health

Short-chain fatty acids (SCFAs), particularly propionate, acetate, and butyrate, are produced through the fermentation of indigestible carbohydrates by gut bacteria [49–52]. While most SCFAs are utilized in the intestine as the source of energy, some are transported to the liver through the hepatic portal vein via the monocarboxylate transporter 1 (MCT-1) and the sodium-coupled monocarboxylate transporter 1 (SMCT-1) receptors [1]. The function of SCFAs in NAFLD is not entirely understood, but elevated fecal SCFA levels and a predominance of SCFA-producing bacteria are observed in NAFLD subjects compared to healthy individuals [53–55]. In general, butyrate primarily provides nourishment to colon epithelial cells, whereas propionate supports gluconeogenesis and cholesterol synthesis, and acetate acts as the key for lipogenesis and cholesterol biosynthesis in the liver [56–59].

Additionally, SCFAs influence liver metabolism, its metabolic substrates, and signaling molecules through various mechanisms, including appetite regulation, energy metabolism,

insulin resistance, and adipose tissue metabolism. They manifest their metabolic effects by binding to G-protein-coupled receptors (GPRs) such as GPR41, GPR43, and GPR109A [51,59,60], with GPR43 being the most crucial for regulating insulin sensitivity and inflammatory responses, thereby influencing fat accumulation and metabolism in the liver cells and adipose tissue [59–61]. They activate GPR41 and GPR43, prompting enteroendocrine cells to release peptide YY (PYY), which slows intestinal transit and enhances nutrient absorption [62,63]. Furthermore, GPR activation stimulates glucagon-like peptide-1 (GLP-1), which promotes hepatic lipid β-oxidation and clinically reduces hepatic steatosis. SCFAs also activate AMP-activated protein kinase (AMPK), stimulating hepatic autophagy to facilitate triglyceride hydrolysis and free fatty acid β-oxidation [54,64,65].

Beyond metabolism, SCFAs play a role in immune modulation by stimulating pro-inflammatory T cells like Th1 and Th17 and influencing inflammatory cytokines, such as TNF-α, a key intermediator of liver inflammation [54,66]. Additionally, systemic acetate can enhance the recall response function of memory CD8+ T cells, thereby improving infection control [67]. Further research is needed to fully understand the characteristics and function of SCFAs and their role in NAFLD pathogenesis, particularly in relation to gut microbiome imbalance and the gut–liver axis.

## 2.1. Bile Acid Metabolism and Circulation Insights

Bile acids (BAs) play a significant role in enterohepatic circulation, particularly in the context of NAFLD. It is synthesized from cholesterol in the liver and secreted into the intestine to aid in lipid digestion and absorption [68,69]. Gut microbiota, especially *Bacteroidetes*, *Lactobacillus*, *Bifidobacterium*, and *Clostridium XIVa* with the help of enzyme bile salt hydrolases (BSH), converts primary BAs into secondary BAs, which are then reabsorbed in the ileum and carried back to the liver via the portal vein [1,49,70,71]. This enterohepatic circulation maintains BA homeostasis and is intricately linked to gut–liver communication [72]. In NAFLD, gut dysbiosis alters BA metabolism, causing an increase in BA synthesis, leading to higher levels of primary BAs and a disrupted primary-to-secondary BA ratio [73]. This imbalance affects the signaling of BA receptors, such as the farnesoid X receptor (FXR) [20,74] and Takeda G-protein receptor 5 (TGR5), which regulate lipid and glucose metabolism, insulin sensitivity, and inflammatory responses [70,75,76]. The FXR, which is primarily located in the liver and intestines, upon activation reduces hepatic lipogenesis and inflammation by inhibiting the NF-κB pathway [71]. Conversely, suppressed FXR activity, which is often observed in NAFLD, leads to increased lipid accumulation and inflammation [77]. For instance, germ-free mice, which lack gut microbiota, are resistant to liver steatosis, highlighting the role of microbial interactions in BA metabolism and NAFLD pathogenesis [78]. TGR5, another important bile acid receptor, upon activation, modulates glucose homeostasis and inflammatory cytokine production, which further improves liver steatosis and hepatocyte damage [79,80]. Dysbiosis-induced alterations in BA levels and signaling further exacerbate NAFLD by promoting hepatic lipid accumulation and reducing insulin sensitivity [80]. Clinical and preclinical studies demonstrate that targeting the intestinal FXR can influence BA metabolism and NAFLD outcomes, although differences between human and mouse models necessitate further research [54]. Maintaining healthy BA metabolism and gut microbiota is essential for preventing NAFLD progression, and understanding the gut–liver axis offers promising avenues for gut microbiota-targeted therapies.

## 2.2. Intestinal Barrier Integrity and Portal Vein Dynamics in the Gut–Liver Axis

Intestinal permeability, crucial for regulating gut–liver communication is determined by the integrity and robustness of the gut intestinal barrier, which comprises the mucus layer, intestinal epithelium, and the gut vascular barrier (GVB) and regulates the transport and circulation from the gut to the liver [81,82]. Enterocytes linked together by tight junction proteins, namely E-cadherins, occludins, and claudins, regulate the ingress into the hepatic portal vein and its way into the liver [81,82]. The portal vein provides a direct link

between the intestine and the liver, delivering approximately 70% of blood, nutrients, and beneficial microbial products like SCFAs to the liver [31,53,83,84]. Gut dysbiosis disrupts the gut barrier integrity, allowing toxic factors like live bacteria, bacterial components (DAMPs, LPS), and proinflammatory metabolites such as ethanol and ammonia to enter the hepatic portal circulation and reach the liver. These toxins further activate immune cells and respective inflammatory pathways, thereby contributing to the NAFLD development and progression [85].

Impaired gut barrier function is a widely recognized characteristic of dysbiosis in subjects with NAFLD and NASH [1,86]. For instance, several studies have reported that the severity of NAFLD is associated with higher zonulin levels in patients compared to healthy controls [87–89]. Furthermore, in a meta-analysis recruiting 128 NAFLD patients, an increase of 39.1% was reported in intestinal gut permeability in NAFLD subjects as compared to 6.8% in healthy controls [61,90]. Clinically, it has been observed that patients with NAFLD exhibit higher portal LPS levels than those with simple steatosis. Additionally, metabolic alterations in the portal system, such as elevated L-tryptophan, DL-3-phenylacetic acid, and glycocholic acid, are observed in NAFLD patients [19,91].

Gut microbiota can enhance intestinal gut integrity by generating and synthesizing metabolites such as SCFAs, which reinforce and strengthen tight junctions [86]. Some bacteria like *Akkermansia muciniphila* can improve gut permeability and affect NAFLD development and progression by regulating tight junctions [83]. However, gut dysbiosis can disrupt these tight junctions, thereby allowing bacteria and their metabolites to translocate from the gut lumen to the liver where Kupffer cells release inflammatory cytokines, exacerbating liver inflammation [84]. This inflammatory response is generated via the stimulation of TLRs on Kupffer cells leading to the subsequent activation of the inflammatory cascade and production of proinflammatory cytokines namely interleukin (IL)-1β, tumor necrosis factor (TNF) α, and interferons [85]. Among the TLR family, TLR2, TRL4, TRL5, and TRL9 are best implicated in the pathogenesis of NAFLD [92]. TLR4 specifically binds with LPS, activates the NF-kB pathway, and further exacerbates the condition by contributing to chronic low-grade inflammation in obesity and NAFLD [20,59,66,93].

### 2.3. Dietary Choline: Implications for the Gut–Liver Axis

Choline, an important component of cell membrane phospholipid [94] is crucial for liver fat metabolism, particularly in forming very low-density lipoproteins (VLDL) necessary for lipid transfer from the liver via the phosphatidylcholine synthase (Pcs) pathway [1,95,96]. According to a study, the high choline intake is directly associated with reduced fatty liver risk in normal-weight women [97]. In animal models of hepatic steatosis, choline deficiency frequently results in liver fat accumulation and gets reversed in patients when choline supplementation is given [93]. The intestinal microbiota can influence choline metabolism by converting dietary choline into trimethylamines, which reduces circulating choline levels. This mimics choline-deficient diets, contributing to NAFLD development by impairing phosphatidylcholine synthesis necessary for VLDL assembly, thereby resulting in triglyceride buildup in hepatocytes [98,99].

Gut commensals such as *E. coli* and *Desulfovibrio desulfuricans* can convert choline to trimethylamine (TMA), which the liver further oxidizes to trimethylamine N-oxide (TMAO), and elevated TMAO levels impair glucose homeostasis, exacerbate insulin resistance, and contribute to atherosclerosis, obesity, and NAFLD [1,98]. A study found that higher plasma TMAO levels are linked to increased all-cause mortality in NAFLD patients [100,101]. Moreover, choline deficiency influences the gut microbiome composition and function, with higher levels of bacteria like *Gammaproteobacteria* and *Erysipelotrichia*, which convert choline to toxic methylamines, being directly connected to liver damage [1,102,103]. This alteration supports the role of gut microbes in NAFLD, with enhanced pathogenic bacterial growth potentially increasing choline demand and contributing to choline depletion and harmful metabolite production [54,103,104]. Thus, the gut microbiome significantly impacts NAFLD through altered choline metabolism and related mechanisms.

## 2.4. Endogenous Ethanol Production and Its Impact on the Gut–Liver Axis

Ethanol is typically produced in very small quantities in the intestine and is metabolized in the liver by the enzyme alcohol dehydrogenases [105]. Hepatic damage in alcoholic liver disease (ALD) and NAFLD is almost identical [106], suggesting a link between blood ethanol levels and changes in gut microbiota [107,108]. Certain gut microbiota can help in the fermentation of dietary carbohydrates and sugars into ethanol, which further gets converted into acetate and acetaldehyde via the upregulation of the CYP2E1 pathway, leading to fatty acid generation and fluctuations in intracellular redox potential, causing an increased intestinal permeability, endotoxemia, and inflammatory response, thus exacerbating liver damage [59,109–111]. Gut dysbiosis can enhance the level of ethanol-producing bacteria such as *Escherichia*, *Klebsiella pneumonia*, *Bacteroides*, *Bifidobacterium*, and *Clostridium*. A study reported a fivefold increase in alcohol-producing bacteria, namely *Escherichia* and *Proteobacteria*, with the quantity of ethanol synthesized to be directly linked to the abundance of *Proteobacteria*, particularly *Klebsiella pneumoniae* [105,112,113]. Additionally, a study identified *Klebsiella pneumonia K14*, a high alcohol producer, as a causative factor for NAFLD [113]. Another study found that mutations in *Klebsiella pneumonia K14* and *Collinsella aerofaciens* influenced alcohol production and were directly correlated with the severity of NAFLD [114,115]. These findings highlight complex interactions between ethanol and NAFLD, warranting further research to clarify ethanol's role in NAFLD development and advancement.

## 2.5. Amino Acid Metabolism in the Gut–Liver Axis

Amino acids and their derived metabolites, including phenylalanine, tryptophan, branched-chain amino acids (BCAAs), and microbiota-derived metabolites, play a crucial role in the development and pathogenesis of NAFLD by influencing liver function through mechanisms such as intestinal integrity, inflammation, lipogenesis, and insulin resistance [116].

Tryptophan, an example of an essential amino acid, is metabolized by gut bacteria through various pathways. The indole pathway converts tryptophan to indole, which enhances intestinal integrity and reduces the severity of NAFLD by promoting tight junction proteins and glycolysis via the PFKFB3 pathway [109,110]. In another pathway, butyrate is catalyzed by indoleamine 2,3-dioxygenase (IDO) to generate kynurenines (Kyn), which further over-activate the Kyn pathway in NAFLD, causing inflammation in various organs [109–117]. Indole-3-acetic acid, a derivative of indole, reduces hepatic lipogenesis and inflammation, thus improving NAFLD [118]. Serotonin, another tryptophan metabolite, inhibits energy expenditure in brown adipose tissue in NAFLD by inhibition of serotonin receptors, namely Thp1 and HTR2a, thereby reducing hepatic steatosis [105,119]. In NAFLD, there is disturbed tryptophan metabolism, and supplementation of tryptophan might increase the intestinal barrier integrity and improve liver NAFLD and function [110]. Phenylalanine and its derivatives, such as phenylacetic acid (PAA), are reported to contribute to the development and pathogenesis of NAFLD [120]. In a study, the transplantation of a liver steatosis-related microbiome into germ-free mice enhanced the PAA and hepatic triglyceride levels, thereby altering the gene expression related to lipogenesis and thus promoting triglyceride accumulation. PAA also inhibits the pAkt pathway, which increases hepatic steatosis by enhancing the use of BCAAs for lipid accumulation [105,121]. BCAAs like Valine, leucine, and isoleucine are linked to NAFLD via the impairment of the TCA cycle in NAFLD [122]. According to a study, elevated BCAA levels are associated with insulin resistance, liver inflammation, and ballooning, indicating severe NAFLD, while free dietary amino acid intake prevents unhealthy metabolic outcomes, as indicated in male mice [105,123,124].

## 2.6. Therapeutic Approaches Targeting NAFLD through Gut Microbiome-Centered Interventions

Gut microbiota interventions show promise in treating NAFLD by modifying microbial composition and function to enhance overall health. This can be achieved through dietary

changes like increasing fiber intake or using probiotics, prebiotics, and postbiotics, as well as through methods such as phage therapy and fecal microbiota transplantation. These approaches aim to restore microbial balance, influence the gut–liver axis, and reduce liver inflammation and fat accumulation while minimizing side effects and promoting a safer, more sustainable management of liver conditions and metabolic health [125].

In contrast, current clinical drugs for NAFLD, such as pioglitazone and vitamin E, primarily target insulin resistance, oxidative stress, and inflammation. Pioglitazone, a thiazolidinedione, improves insulin sensitivity and reduces liver fat content but is associated with adverse effects like weight gain and an increased risk of heart failure. Vitamin E, an antioxidant, helps reduce oxidative stress and inflammation in the liver, showing beneficial effects in non-diabetic NAFLD patients but with potential long-term safety concerns [126].

Other pharmacological agents, such as glucagon-like peptide-1 (GLP-1) agonists and sodium-glucose co-transporter-2 (SGLT2) inhibitors, show promise for improving metabolic parameters and reducing liver fat [127,128]. However, these drugs can have side effects and do not address the underlying gut–liver axis, requiring long-term efficacy and safety evaluation [129,130]. For instance, pioglitazone can cause weight gain and increase the risk of congestive heart failure, metformin carries a risk of lactic acidosis, and vitamin E, while reducing oxidative stress, has been linked to a higher risk of prostate cancer with long-term use [131,132]. Moreover, these drugs often target specific pathways, which may limit their effectiveness in addressing the multifactorial nature of NAFLD [130].

Furthermore, gut microbiota therapies generally exhibit a favorable safety profile with minimal side effects. Most probiotics and prebiotics are well-tolerated, with occasional mild gastrointestinal symptoms such as bloating or gas being the most common adverse effects. Unlike clinical drugs, gut microbiota interventions do not carry risks of severe complications like congestive heart failure, lactic acidosis, or cancer. This makes them a safer option, especially for patients with NAFLD who are at increased risk of cardiovascular events or progressive liver disease [44,133]. Therefore, gut microbiota interventions offer a complementary approach by targeting the root cause of dysbiosis and providing a more holistic treatment option for NAFLD.

## 3. Probiotics

Probiotics are non-pathogenic live microorganisms that are generally considered beneficial for maintaining gut health and are mostly used in diarrhea [134–136] and malnutrition [137,138]. Recently, probiotic supplements have been reported to have worthwhile effects on both intro-intestinal and extro-intestinal diseases, including NAFLD [139,140].

Probiotics can restore gut microbiota balance [141], enhance lipid and glucose profiles [142,143], maintain intestinal gut barrier integrity [144], reduce inflammation [145], and inhibit oxidative stress [146]. These mechanisms theoretically contribute to the effective prevention of NAFLD.

Multiple animal studies have demonstrated that probiotics benefit NAFLD by reducing inflammation, hepatic triglyceride levels, overall body weight, and visceral adipose tissue weight, as well as by improving glucose homeostasis [147]. However, clinical evidence regarding the direct impact of probiotics on NAFLD is still insufficient, despite numerous animal studies exploring their therapeutic mechanisms. In a study aimed to evaluate the overall efficacy of probiotics in treating NASH using a hepatocyte-specific PTEN knockout mouse model which closely resembles human NAFLD indicated that probiotics significantly reduced serum transaminase levels, NAFLD activity scores, and the expression of pro-inflammatory cytokine genes. Additionally, probiotics alleviated oxidative stress, evidenced by anti-oxidative stress markers, and altered glutathione levels, suggesting a potential mechanism of action for their beneficial effects. Overall, probiotics demonstrated positive effects in mitigating NAFLD and preventing carcinogenesis in the PTEN knockout mice model [15]. Another study found that probiotic supplementation with *Lactobacillus acidophilus* in rabbits protected them against NAFLD. The treatment restored liver function, lipolytic gene expression, and antioxidant levels to normal [148].

Another meta-analysis aimed to summarize randomized controlled trials that examined the effects and efficacy of probiotics on NAFLD, which included the assessment of the impact of probiotics on liver function tests, specifically alanine aminotransferase (ALT), aspartate aminotransferase (AST), and gamma-glutamyl transferase (GGT). Overall, probiotics have shown beneficial effects and could be considered as an additional therapeutic approach for managing NAFLD [149]. A similar meta-analysis of 21 randomized clinical trials containing 1037 NAFLD subjects found that probiotic intervention significantly improved liver function tests, blood lipid levels, and blood glucose and insulin levels, thereby reducing hepatic steatosis. However, it did not affect the BMI index, level of inflammatory factors, or insulin resistance significantly. Subgroup analysis revealed that treatments lasting 12 weeks or longer resulted in better improvements in ALT, GGT, triglycerides, and blood sugar. Overall, the study indicates that probiotics effectively regulate liver function, liver steatosis, glucose homeostasis, and blood lipid levels in NAFLD subjects [150].

Over the past decade, clinical investigations into probiotic therapies for NAFLD have shown improvements, but the results and benefits remain contentious. The discrepancies in the trials mentioned above may be attributed to variations in study design, probiotic dosage, types of strains, and supplement duration, as well as the characteristics of the participants. Overall, probiotics may be beneficial in reducing transaminase levels and partially improving lipid profiles, particularly in the initial stages of NAFLD development [151]. However, long-term research is needed to establish more robust evidence and to determine whether probiotics can prevent the advancement and progression of NAFLD to liver cirrhosis and HCC. Currently, there are significant challenges in using probiotics to treat NAFLD. First, selecting the appropriate probiotics is complex due to the diverse range of strains, with different benefits and effects attributed to different strains; multi-strain probiotics have shown more effectiveness than single strains. Second, while initial-stage NAFLD patients have shown improvements with reduced transaminase levels following probiotics treatment, it is unclear whether probiotics can prevent advancement to liver fibrosis. Third, both the study duration and probiotic dosage impact efficacy outcomes. Despite these challenges, probiotics are considered a promising therapeutic strategy and are anticipated to become an effective, significant, and widely used treatment for NAFLD patients [151].

## 4. Prebiotics

Prebiotics are non-viable dietary components linked with gut microbiota modulation and can provide health-related benefits to the host. They primarily include polysaccharides like inulin, cellulose, hemicellulose, resistant starch, and pectins as well as oligosaccharides such as fructooligosaccharides, isomaltooligosaccharides, galactooligosaccharides, lactulose, xylooligosaccharides, and soy oligosaccharides, with fructooligosaccharides being widely researched in NAFLD [152]. These substances selectively stimulate the growth and activity of gut microbiota. Numerous animal studies have demonstrated that prebiotic supplementation can reduce fatty acid synthesis pathways, potentially lowering hepatic triglyceride accumulation caused by fructose. This effect is achieved through reduced expression of enzymes involved in hepatic lipogenesis, including fatty acid synthase and acetyl Co-A carboxylase. Additionally, oligofructose modifies the gut microbiome to promote *Bifidobacterium*, which enhances intestinal barrier function and lowers endotoxin levels [153].

Additionally, in another study on mice models, the administration of prebiotics decreased liver inflammation in obese mice via the glucagon-like peptide-2-dependent pathway, which also positively affected the gut barrier [154]. The study also evaluated the liver-protective effects of COS23, an enzymatically catalyzed byproduct of chitosan oligosaccharide (COS), in diet-induced obese mice and reported that it significantly reduced hepatic steatosis and improved liver injury by decreasing toxic lipids such as triglycerides and free fatty acids in the liver. It also regulated lipid-related pathways and inflammation while altering lipid profiles in plasma. Additionally, COS23 modulated gut microbiota, decreas-

ing *Mucispirillum* and increasing *Coprococcus* abundance, and improved intestinal barrier function. These findings suggest that COS23 holds promise as a clinical treatment for NAFLD. However, human clinical trials are essential to corroborate this information [155].

Despite limitations in many studies, such as small sample sizes and insufficient data on patients' diets and lifestyles, treatments with probiotics and prebiotics for NAFLD show promise. Strong evidence from short-term, high-quality human studies supports using dietary probiotics and prebiotics as a possible therapeutic approach for NAFLD. Nevertheless, additional research is required to establish a link between these results and alterations in gut microbiota [156].

## 5. Postbiotics

Postbiotics, which contain various bioactive substances, might exert promising effects by reducing hepatic lipid build-up [44,157]. A study explored the preventive role of postbiotics derived from *Lactobacillus paracasei* on NAFLD. The findings revealed that simultaneous ingestion of a high-fat diet (HFD) and postbiotics slowed weight gain, suppressed epididymal white fat hypertrophy and glycemic hike, improved serum biochemical markers associated with blood lipid metabolism, and reduced hepatic steatosis and low-grade liver inflammation in mice [158]. Bacterial sequencing demonstrated that postbiotics altered the gut microbiota in HFD mice, elevating the levels of *Akkermansia* and decreasing the relative abundance of the *Lachnospiraceae NK4A136* group, *Bilophila,* and *Ruminiclostridium* [158].

## 6. Fecal Microbiota Transplantation (FMT)

FMT involves transferring fecal matter from a healthy donor to a patient, which can potentially correct dysbiosis and improve liver health. This procedure helps restore the balance of commensal gut microbiota and enhances the gut's natural defenses, preventing the entry and translocation of potential pathogens [159]. As a result, FMT has been proposed to address other dysbiosis-related conditions in the gut microbiome, including those observed in NAFLD [48]. FMT restores the microbiome-mediated regulation of gut barrier integrity, preventing the entry and migration of potential pathogens, such as *Clostridiales* [160]. Initial animal studies provided the first evidence of FMT's impact on NAFLD. Leroy et al. found that FMT from NAFLD mice triggered NAFLD development in most recipient mice, underscoring the substantial colonization of specific bacteria following FMT [161]. Additionally, a study investigated two different groups of mice fed a high-fat diet, with one group receiving FMT from healthy donors. This intervention led to a notable decrease in typical histological features associated with NAFLD in the treated group [162]. Interestingly, improvements in NASH-related histological features, including liver fibrosis and inflammatory infiltrates, were also observed post-FMT. These improvements correlated with reductions in body weight, fat content, and serum transaminase levels [163]. The clinical trials reported have demonstrated promising results of FMT in NAFLD, benefiting both diabetic and non-diabetic patients, with improvements noted in glycemic control and liver steatosis. Furthermore, initial reports suggest FMT may also be safe for use in patients with liver cirrhosis, a progression of NASH [48,164].

FMT seems to be a safe and effective treatment for NAFLD, though further randomized controlled trials (RCTs) and long-term follow-up studies are necessary to fully evaluate its effects and efficacy, especially in lean NAFLD patients who often do not benefit significantly from lifestyle and dietary changes, cholesterol-lowering agents, or probiotics [44].

## 7. Phage Therapy

Phage therapy, long overshadowed by antibiotics, is now being reconsidered as a potent tool against antibiotic-resistant bacteria [165]. Given the mechanistic links between bacterial microbiota, gastrointestinal, and liver disorders, the targeted manipulation of the gut microbiota through phages' selective bactericidal action has garnered renewed attention. The human gut virome, dominated by phages, shows significant alterations in patients with liver diseases when compared to healthy controls [125]. Changes in the

phageome have been observed in NAFLD patients, with lower viral diversity in advanced stages and higher abundance of *Enterobacteria*, *Escherichia*, and *Lactobacillus phages* [125].

Phages can target and reduce specific pathobionts, as evidenced by preclinical studies showing that selective elimination of pathobionts via phages improves inflammatory bowel disease, ethanol-induced liver disease, primary sclerosing cholangitis, and NAFLD in mouse models [166]. For example, phages targeting the cytolysin-producing *Enterococcus faecalis* reduced the circulation and presence of harmful protein cytolysin, which is linked to severe liver disease and mortality in NALD subjects [167]. In studies with humanized mice, oral administration of phages against cytolysin-positive *E. faecalis* reduced ethanol-induced liver injury, liver steatosis, and liver and gut inflammation. These findings highlight the role of harmful bacteria in fatty liver diseases and demonstrate that phage therapy can reduce liver disease and its associated complications in preclinical models. Phages typically have a narrow host range, infecting closely related strains within species, which minimizes collateral damage to the recipient's microbiome but may also limit their ability to broadly modulate the gut microbiome [125]. The impact of the gut intestinal virome on bacterial microbiota and NAFLD progression remains largely unclear. However, phage therapy shows potential for treating NAFLD, as demonstrated by the prevention of NAFLD development in mice after eliminating ethanol-producing *Klebsiella pneumoniae* through phage therapy [113,168].

## 8. Conclusions and Future Prospects

The gut–liver axis plays a fundamental role in the development and pathogenesis of NAFLD. Gut dysbiosis, along with increased intestinal permeability, and altered bile acid signaling are the key factors responsible for these liver diseases. Advances in understanding this axis are driving the development of microbiota-based targeted therapeutic tools to prevent and treat NAFLD, ultimately improving patient outcomes. Despite the proven therapeutic potential of probiotics, fecal microbiota transplantation, and phage therapy in these conditions, comprehensive safety assessments and data on microbiota–host interactions are lacking. Most of the current knowledge is derived from animal studies, which face translational limitations due to physiological differences between species. Hence, large-scale controlled human studies with standardized and the most effective strains are needed to optimize dosages and treatment durations to individualize therapy for better disease management. The findings then need to be translated into clinical practice. Comprehensive studies on microbiota–host interactions will help pinpoint specific microbial strains or metabolites critical to NAFLD pathogenesis, aiding in the identification of reliable biomarkers for early diagnosis and monitoring. Additionally, integrating multi-omics technologies will offer a deeper, more comprehensive understanding of the molecular mechanisms at play, paving the way for personalized medicine and novel therapeutic targets. By advancing research in these areas, we can develop innovative strategies to mitigate the global burden of NAFLD and enhance patient outcomes.

**Author Contributions:** Conceptualization, M.K. (Manoj Kumar) and S.S.; writing and original draft preparation, M.K. (Mona Kriti) and S.S.; review and editing, R.C., F.M., M.M., A.J., V.V. and R.N.; visualization R.T. All authors have read and agreed to the published version of the manuscript.

**Funding:** This research received no external funding.

**Institutional Review Board Statement:** Not applicable.

**Informed Consent Statement:** Not applicable.

**Data Availability Statement:** No new data were created or analyzed in this study. Data sharing is not applicable to this article.

**Conflicts of Interest:** The authors declare no conflicts of interest.

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
