# Peer review of "Deciphering the Gut–Liver Axis: A Comprehensive Scientific Review of Non-Alcoholic Fatty Liver Disease"

_livers, doi:10.3390/livers4030032_

Round 1
Reviewer 1 Report
Comments and Suggestions for Authors
This manuscript is very interesting, focusing on the use of microbiota to treat NAFLD.
I suggest strengthening the explanation of the mechanisms through which microbiota affects NAFLD. Please clarify the mechanisms by which microbiota treats NAFLD and compare these with the mechanisms and effects of current clinical drugs. The results demonstrating efficacy in animal experiments should be discussed in greater depth.
Author Response
Comments: This manuscript is very interesting, focusing on the use of microbiota to treat NAFLD. I suggest strengthening the explanation of the mechanisms through which microbiota affects NAFLD. Please clarify the mechanisms by which microbiota treats NAFLD and compare these with the mechanisms and effects of current clinical drugs. The results demonstrating efficacy in animal experiments should be discussed in greater depth.
Response: Thank you for your positive feedback and valuable suggestions. We have already written the mechanisms through which microbiota affects NAFLD, specifically, we have added detailed descriptions of the pathways involved, such as the modulation of bile acids, short-chain fatty acids, inflammatory cytokines and the use of probiotics that contains microorganisms beneficial to humans. We have also included a comparison of these mechanisms with those of current clinical drugs used to treat NAFLD in the “Therapeutic Approaches Targeting NAFLD through Gut Microbiome-Centered Interventions” sub-section. Furthermore, the discussion on the efficacy of microbiota in animal experiments has been expanded to provide a deeper understanding of the results. These revisions have been highlighted bold in the manuscript for easy identification.
Reviewer 2 Report
Comments and Suggestions for Authors
The authors reviewed the mechanism of the gut-liver axis and the pathogenesis of metabolic-associated steatotic liver disease (according to the new terminology). The review is well-written and well-organized. Still, as the new terminology has been in place for one year now, the authors probably must consider using the new nomenclature in their paper. Also, they should present the different categories of the newly named steatotic liver disease.
The paper needs a thorough check for the use of abbreviated swords. The authors should avoid defining one abbreviated word twice and use form that form in all papers. The figure should enlarged to be easily understandable.
The sections of the paper must be organized better with their subtitles.
Author Response
Comments: The authors reviewed the mechanism of the gut-liver axis and the pathogenesis of metabolic-associated steatotic liver disease (according to the new terminology). The review is well-written and well-organized. Still, as the new terminology has been in place for one year now, the authors probably must consider using the new nomenclature in their paper. Also, they should present the different categories of the newly named steatotic liver disease.
Response: Thank you for your insightful comments. we already had discussed about the new nomenclature system in the introduction part of the paper. Since most of the paper is written with old terminologies, we have retained the old terminologies to maintain consistency and avoid conflicts within the text. Also, the different categories of the newly named steatotic liver disease relevant to this paper such as MASLD, MASH and SLD have already been discussed. Additionally, we have added further recent information regarding the same which have been highlighted bold for clear identification.
Comments: The paper needs a thorough check for the use of abbreviated swords. The authors should avoid defining one abbreviated word twice and use form that form in all papers.
Response: We have thoroughly reviewed the manuscript to ensure that each abbreviated term is defined only once and used consistently throughout the text. This revision has been carefully highlighted in the manuscript.
Comments: The figure should enlarged to be easily understandable.
Response: We have enlarged the figures in the manuscript to improve clarity and readability. The revised figures are now more detailed and easier to understand.
Comments: The sections of the paper must be organized better with their subtitles.
Response: We have reorganized the sections of the manuscript with clear and descriptive subtitles to enhance the overall structure and readability. This reorganization helps to guide the reader through the content more effectively.
Reviewer 3 Report
Comments and Suggestions for Authors
There are some comments.
1. In the title, it would be better to modify 'NAFLD' to 'nonalcoholic fatty liver disease'.
2. Please insert 'Figure 2' in the text.
3. It would be better to explain progression mechanism of NAFLD to HCC in detail.
4. It would be better to replace Figure 1, 2, and 3 with more comprehensive and clearer photos with clearly legible labels and descriptions.
5. Please modify the reference format according to the Livers Journal guideline.
6. Please italicize the microorganism names.
For example, Firmicutes, Bacteroidetes
Comments on the Quality of English Language
Please check English grammar.
For example, Non-alcoholic fatty liver disease (NAFLD) have ->
Non-alcoholic fatty liver disease (NAFLD) has
Author Response
Comments: In the title, it would be better to modify 'NAFLD' to 'non-alcoholic fatty liver disease'.
Response: Thank you for your suggestion. We have modified the title to replace 'NAFLD' with 'non-alcoholic fatty liver disease' for better clarity and comprehensibility.
Comments: Please insert 'Figure 2' in the text.
Response: 'Figure 2' has now been correctly inserted into the text at the appropriate location.
Comments: It would be better to explain progression mechanism of NAFLD to HCC in detail.
Response: We have expanded the introduction part of the manuscript to include a detailed explanation of the progression mechanisms from non-alcoholic fatty liver disease (NAFLD) to hepatocellular carcinoma (HCC). This section now discusses the key molecular and cellular processes involved, providing a comprehensive overview of the disease progression.
Comments: It would be better to replace Figure 1, 2, and 3 with more comprehensive and clearer photos with clearly legible labels and descriptions.
Response: We have replaced Figures 1, 2, and 3 with more comprehensive images that have improved clarity and visual impact, aiding in better understanding of the content.
Comments: Please modify the reference format according to the Livers Journal guideline.
Response: We have revised the reference section to conform to the Livers Journal guidelines. All references are now correctly formatted as per the journal's requirements.
Comments: Please italicize the microorganism names. For example, Firmicutes, Bacteroidetes
Response: We have reviewed the manuscript to ensure that all microorganism names are italicized correctly. These revisions have been highlighted in bold.
Comments on the Quality of English Language
Comments: Please check English grammar.
For example, Non-alcoholic fatty liver disease (NAFLD) have ->
Non-alcoholic fatty liver disease (NAFLD) has
Response: We have thoroughly reviewed the manuscript for grammatical and syntactical issues to improve the overall quality of English.
Reviewer 4 Report
Comments and Suggestions for Authors
Dear Authors
Thank you for this review of the state -of -art concerning gut liver axis in steatotic liver diseases
I consider it an adequate review, however, the manuscript should be revised according to the current definitions of hepatic steatotic disease. I attach the text for minor revisions and draw attention to references that are incompleteKindest regards

Minor corrections
Author Response
Comment: I consider it an adequate review, however, the manuscript should be revised according to the current definitions of hepatic steatotic disease. I attach the text for minor revisions and draw attention to references that are incomplete.
Response: Thank you for your feedback. I have revised the manuscript according to the current definitions of hepatic steatotic disease and made the necessary minor revisions. All references have been updated and managed via EndNote.
Round 2
Reviewer 2 Report
Comments and Suggestions for Authors
The authors addressed my comments, including the one related to the new terminology. Even though I still consider that the new terminology should be adopted, the authors explained their reason.
Still, some minor edits must be made related to the abbreviated words used in the Figures.
Author Response
Comments:
The authors addressed my comments, including the one related to the new terminology. Even
though I still consider that the new terminology should be adopted, the authors explained
their reason. Still, some minor edits must be made related to the abbreviated words used in
the Figures.
Response:
We appreciate the reviewer’s understanding regarding our decision to retain the current
terminology. We have provided our rationale for this choice and believe it is justified within
the context of our manuscript. Additionally, we have reviewed all figures and ensured
consistency in the use of abbreviated terms. Please find the revised figures attached and all
the changes highlighted bold.
Reviewer 3 Report
Comments and Suggestions for Authors
The manuscript was well-revised. However, there are some minor comments to address:
It would be better to unify "NAFLD" as either "non-alcoholic fatty liver disease" or "nonalcoholic fatty liver disease" throughout the manuscript.
Please ensure the figure on page 8 is removed.
Please correct "E. coli" to italics in Figure 2.
Please ensure the figure on page 11 is removed.
Please correct "hepatocarcinoma" to "hepatocellular carcinoma" in Figure 3.
Please confirm that the reference format adheres to the Livers Journal guidelines.
Please leave a space when adding references.
Comments on the Quality of English LanguagePlease check English grammar.
Author Response
The manuscript was well-revised. However, there are some minor comments to address:
Comments:
It would be better to unify "NAFLD" as either "non-alcoholic fatty liver disease" or
"nonalcoholic fatty liver disease" throughout the manuscript.
Response:
We have standardized the term "non-alcoholic fatty liver disease" throughout the manuscript
for consistency.
Comments:
Please ensure the figure on page 8 is removed.
Response:
The figure on page 8 has been removed and added at the end of the paper before declaration
section.
Comments:
Please correct "E. coli" to italics in Figure 2.
Response:
We have corrected "E. coli" to italics in Figure 2.
Comments:
Please ensure the figure on page 11 is removed.
Response:
The figure on page 11 has been removed and added at the end of the paper before declaration
section.
Comments:
Please correct "hepatocarcinoma" to "hepatocellular carcinoma" in Figure 3.
Response:
We have corrected "hepatocarcinoma" to "hepatocellular carcinoma" in Figure 3.
Comments:
Please confirm that the reference format adheres to the Livers Journal guidelines.
Response:
We have reviewed the references according to the Livers Journal guidelines.
Comments:
Please leave a space when adding references.
Response:
Spaces have been added between references as requested.